

**Vegetation dynamics and climate seasonality jointly control the interannual**
**catchment water balance in the Loess Plateau under the Budyko framework**
**Tingting Ning[1,2], Zhi Li[3], and Wenzhao Liu[1,2]**
[1] *State Key Laboratory of Soil Erosion and Dryland Farming on the Loess Plateau, Institute of Soil and Water*
*Conservation, CAS & MWR, Yangling, Shaanxi 712100, China*
[2] *University of the Chinese Academy of Sciences, Beijing 100049, China*
[3] *College of Natural Resources and Environment, Northwest A&F University, Yangling, Shaanxi 712100, China*
*Correspondence to: Wenzhao Liu, wenzhaoliu@hotmail.com*
**Abstract**. Within the Budyko framework, the controlling parameter ($\omega$ in the Fu equation) is widely
considered to represent landscape conditions in terms of vegetation coverage ($M$); however, some
qualitative studies have concluded that climate seasonality ($S$) should be incorporated in $\omega$. Here, we
discuss the relationship between $\omega$, $M$, and $S$, and further develop an empirical equation so that the
contributions from $M$ to actual evapotranspiration ($ET$) can be determined more accurately. Taking 13
catchments in the Loess Plateau as examples, $\omega$ was found to be well correlated with $M$ and $S$. The
developed empirical formula for $\omega$ calculations at the annual scale performed well for estimating $ET$ by
the cross-validation approach. By combining the Budyko framework with the semi-empirical formula,
the contributions of changes in $\omega$ to $ET$ variations were further decomposed as those of $M$ and $S$.
Results showed that the contributions of $S$ to $ET$ changes ranged from 0.1% to 65.6% (absolute values);
therefore, the impacts of climate seasonality on $ET$ cannot be ignored. Otherwise, the contribution of $M$
to $ET$ changes will be estimated with a large error. The developed empirical formula between $\omega$, $M$, and
$S$ provides an effective method to separate the contributions of $M$ and $S$ to $ET$ changes.
KEYWORDS: Budyko framework; Controlling parameter; Vegetation dynamics; Climate seasonality;
Loess Plateau



# 1. Introduction

The water cycle has been influenced greatly by human activities and climate change since the 1960s, and considerable variability in hydrological processes has been observed in many basins around the world; this has led to a series of problems concerning essential water resources (Stocker et al., 2014). Analyses of the mechanisms of the interactions among the water balance, climate, and catchment surface conditions are important for understanding these complex processes at different spatio-temporal scales (Zhang et al., 2008), and such work has practical significance in regard to the improvement of water resources and land management (Rodriguez-Iturbe, 2000; Xu et al., 2014).

Budyko (1948, 1974) postulated that precipitation ($P$, represents the water supply from the atmosphere) and potential evapotranspiration ($ET_0$, represents the demand by the atmosphere) are the two dominant variables that control the long-term average water balance. The Budyko framework is considered one of the most abiding frameworks linking climatic conditions to the runoff ($R$) and actual evapotranspiration ($ET$) of a catchment (Donohue et al., 2007), and it has been used successfully to investigate interactions between hydrological processes, climate variability, and landscape characteristics (e.g. (Milly, 1994; Woods, 2003; Yokoo et al., 2008; Yang et al., 2009)). A series of empirical formulas have been developed for the Budyko curve based on theoretical research and case studies of regional water balance over the past 50 years. Among them, the Fu (Fu, 1981; Zhang et al., 2004) and Choudhury–Yang equations (Choudhury, 1999; Yang et al., 2008) have been used widely; furthermore, the controlling parameters $\omega$ (in the Fu equation) and $n$ (in the Choudhury–Yang equation) are related linearly (Yang et al., 2008).

Deviations from the Budyko curve have been detected in previous studies, which indicates that in addition to climate conditions, other variables can also influence the variability of regional water balances (Yang et al., 2007; Wang and Alimohammadi, 2012). Two kinds of factors have been identified to be responsible for the deviations. The first type of factors are related to land surface conditions, and



these include vegetation dynamics (Donohue et al., 2007; Yang et al., 2009; Donohue et al., 2010; Li et al., 2013; Zhang et al., 2016), soil properties, and topography (Yang et al., 2007; Peel et al., 2010). The second type of factors include seasonal climate variability (in addition to $P$ and $ET_0$), such as storm depth (Shao et al., 2012; Li, 2014), frequency of daily rainfall (Milly, 1994), and differences in the timing of $P$ and $ET_0$ (Budyko, 1961; Potter et al., 2005). All of these factors can be encoded into the controlling parameter of the Budyko equations (e.g. $\omega$ in the Fu equation and $n$ in the Choudhury–Yang equation). So far, a great deal of attention has been paid to the relationships between land surface conditions and the controlling parameter. Based on satellite products of vegetation such as the Normalized Difference Vegetation Index (NDVI), vegetation has been found to correlate well with the controlling parameter, and some empirical relationships have been successfully developed (Yang et al., 2009; Li et al., 2013). In particular, the controlling parameter can be better represented by vegetation when higher spatiotemporal resolution products are used. Therefore, the impacts of dynamic changes in vegetation on hydrology can be effectively quantified.

Many current studies attribute any effects of the controlling parameter to landscape characteristics (Roderick and Farquhar, 2011; Zhou et al., 2015; Zhang et al., 2016). However, both empirical evidence and modelling tests have demonstrated the important function of climate seasonality on catchment water yield, and thereby, evidence exists that climate seasonality also strongly affects the controlling parameter in the Budyko equations (Berghuijs and Woods, 2016). Some indices and models have thus been developed to address this issue, and several potential solutions have been discussed (Milly, 1993, 1994; Potter et al., 2005; Yokoo et al., 2008; Feng et al., 2012; Li, 2014). So far, two notable advances related to this problem have come from Yang et al. (2012) and Jiang et al. (2015). Yang et al. (2012) introduced the climate seasonality index into the Budyko framework and proposed an empirical equation to include its effect in the estimation of the long-term controlling parameters; however, by focusing on the mean annual scale, the effects of vegetation dynamics were not considered. Jiang et al. (2015) proposed an empirical formula for the parameter $\omega$ with the factors of climate (represented by





temperature and $ET_0$) and human activities, and they found that the performance was very good ($R^2 >$
0.9); however, as parameter ω derived from the water balance was actually a function of $ET_0$, a
self-correlation phenomenon may exist in their formula. Therefore, how the vegetation dynamics and
climate seasonality jointly control the interannual variability in the controlling parameters needs further
interpretation.
Therefore, the primary motivation behind this study was to detect the potential linkages between
the controlling parameter and surface condition change, as well as climate seasonality at an annual scale.
The specific objectives were to derive an appropriate analytic formula between parameter ω in the Fu
equation and the above two factors for typical catchments in the Loess Plateau, and then, quantify the
impacts of vegetation change and climate seasonality variability on the catchment water balance.

## 2. Methods

### 2.1. Annual water balance definition

The Budyko framework assumes that the long-term average water balance is in a steady state
(Wang and Alimohammadi, 2012), and the water storage change ($\Delta S$) in a catchment can be negligible.
The interannual variability of the water balance in individual basins can also be studied by overlooking
the interannual variation of the catchment water storage ($\Delta S$) (Sankarasubramanian and Vogel, 2002;
Yang et al., 2007; Potter and Zhang, 2009). However, water storage change can be great when analysing
the interannual variability of the water balance (Wang, 2012). To minimize the potential errors
introduced by neglecting water storage variation, the hydrological year (Sivapalan et al., 2011; Carmona
et al., 2014) and moving windows (Jiang et al., 2015) were introduced to the time series of annual
hydrological variables. Similar to Sivapalan et al. (2011) and Carmona et al. (2014), the hydrological
year rather than the calendar year is introduced to calculate the annual $ET$, and this is called the
''measured'' $ET$ in the subsequent discussion. Specifically, as the study area has a semiarid climate with




most rainfall occurring in summer and autumn (July–September), a hydrological year is defined as July
to June of the following year. In this way, the water input occurs mainly at the beginning of the year and
the water is consumed within that year.

**2.2. Identification of factors determining parameter $\omega$ in Fu's equation**

The Fu equation is used in this study with the following expressions:
$$\frac{ET}{P} = 1 + \frac{ET_0}{P} - \left[1 + \left(\frac{ET_0}{P}\right)^{\omega}\right]^{1/\omega} \text{ or}$$
$$\frac{ET}{ET_0} = 1 + \frac{P}{ET_0} - \left[1 + \left(\frac{P}{ET_0}\right)^{\omega}\right]^{1/\omega} \tag{1}$$
where $\omega$ is the controlling parameter of the Budyko curve. $ET_0$ is calculated by using the equation of
Priestley and Taylor (1972).
The important issue regarding the parameterization of $\omega$ in Fu's equation is to choose factors with
physical meanings. According to the results from related studies, land surface conditions can be mainly
represented by vegetation, which was also true in this study. With an arid to semiarid climate, water
availability is the key factor that controls vegetation dynamics. Although soil properties and topography
also influence vegetation growth, their impacts can be ignored on an annual scale because they would
be expected to be almost constant over a year. Therefore, vegetation dynamics (i.e. vegetation coverage)
were chosen to represent the variations in surface conditions. The vegetation coverage ($M$) was
estimated by the following equation (Yang et al. (2009)):
$$M = \frac{NDVI - NDVI_{min}}{NDVI_{max} - NDVI_{min}} \tag{2}$$
where $NDVI_{max}$ and $NDVI_{min}$ are the NDVI values of dense forest (0.80) and bare soil (0.05),
respectively.
Two limiting conditions were used to illustrate the effects of seasonal variations in coupled water
and energy on the regional water balance. If $P$ and $ET_0$ are in phase, the intra-annual distribution of





precipitation is very symmetrical, and thus, $R \to 0$ in non-humid regions and $ET \to P$. However, if $P$
and $ET_0$ are out phase, the total precipitation of one year is concentrated at a certain moment, and thus,
$R \to P$ and $ET \to 0$. Therefore, the impacts of seasonal variations in coupled water and energy on the
regional water balance cannot be neglected, and they can only be reflected by the controlling parameter.
In this study, the climate seasonality index ($S$), as introduced by Milly (1994) and Woods (2003), was
used to reflect the non-uniformity in the annual distribution of water and heat:
$$S = \left| \delta_P - \delta_{ET_0} \emptyset \right| \tag{3}$$
where $\delta_P$ and $\delta_{ET_0}$ are the seasonal amplitude of precipitation and potential evapotranspiration,
respectively, which can be expressed by sine functions and fitted by the monthly $P$ and $ET_0$ values. $\emptyset$
is the dryness index.
**2.3. Evaluating the contributions of climate change and surface condition alterations to *ET* trends**
Based on the climate elasticity method, which was introduced by (Schaake and Waggoner, 1990)
and improved by (Sankarasubramanian et al., 2001), the contribution of change for each climate factor
to pan evaporation and/or $ET_0$ was defined as the product of the partial derivative and slope of the trend
for the climate factor (Zheng et al., 2009). Here, we extend this method to the Budyko equation and
incorporate the vegetation coverage and climate seasonality.
The contributions of $P$, $ET_0$, and ω changes to the $ET$ trends can be assessed by taking the
derivative of Eq. (1) with respect to time:
$$\frac{\mathrm{d}ET}{dt} = \frac{\partial f}{\partial P}\frac{dP}{dt} + \frac{\partial f}{\partial ET_0}\frac{dET_0}{dt} + \frac{\partial f}{\partial \omega}\frac{d\omega}{dt} \tag{4}$$
In order to simplify the calculation process for the partial derivative in Eq. (4), the multiple linear
regression method is applied in our study by using the time series of $ET$, $P$, $ET_0$, and ω for each basin,
and the regression coefficients approximately replace the partial derivative. Thus, Eq. (4) can be



rewritten as:
$$\frac{dET}{dt} = A * \frac{dP}{dt} + B * \frac{dET_0}{dt} + C * \frac{d\omega}{dt} \qquad (5)$$
where A, B, and C are the regression coefficients. Alternatively, it can be expressed as:
$$L\_(ET) = C\_(P) + C\_(ET_0) + C\_(\omega) \qquad (6)$$
where L_(ET) is the slope of *ET*, and C_(P), C_(ET_0), and C_(ω) are the respective contributions of *P*,
*ET_0*, and ω to the *ET* trends.
After obtaining the contribution of parameter ω to the *ET* change, the contributions of vegetation
coverage (*M*) and climate seasonality (*S*) to *ET* change can be further decomposed as follows.
First, the contributions of *M* and *S* to parameter ω are calculated by using an equation similar to
Eq. (4) and the partial derivatives are replaced by the regression coefficients for the relationships
between ω and *M* as well as *S*. Furthermore, the individual relative contributions (RC) of *M* and *S* to
ω can be calculated. Then, the contributions of *M* (C_(M)) and *S* (C_(S)) to *ET* trends can be obtained
as follows:
$$C\_(M) = C\_(\omega) \times RC\_(M) \qquad (7a)$$
$$C\_(S) = C\_(\omega) \times RC\_(S) \qquad (7b)$$
**3. Study area and data**
The Loess Plateau, which is located in the middle reaches of the Yellow River in China,
experiences a sub-humid and semiarid continental monsoon climate (Liang et al., 2015). Frequent heavy
summer storms, sparse vegetation coverage, easily erodible wind-deposited loess soil, and a long
agricultural history have all contributed to severe drought and soil erosion problems in this region (Li et
al., 2012). To recover and preserve the ecosystem, the Chinese government has launched numerous soil
and conservation measures since the 1950s, and these include biologic measures ("Grain to Green"
Project) and engineering measures (building terraces and sediment trapping dams) (Mu et al., 2007). As



a result, the hydrological processes of this area have undergone significant changes (Huang and Zhang, 2004; Zhang et al., 2008). Thirteen catchments on the Loess Plateau were selected as our study area (Figure 1).

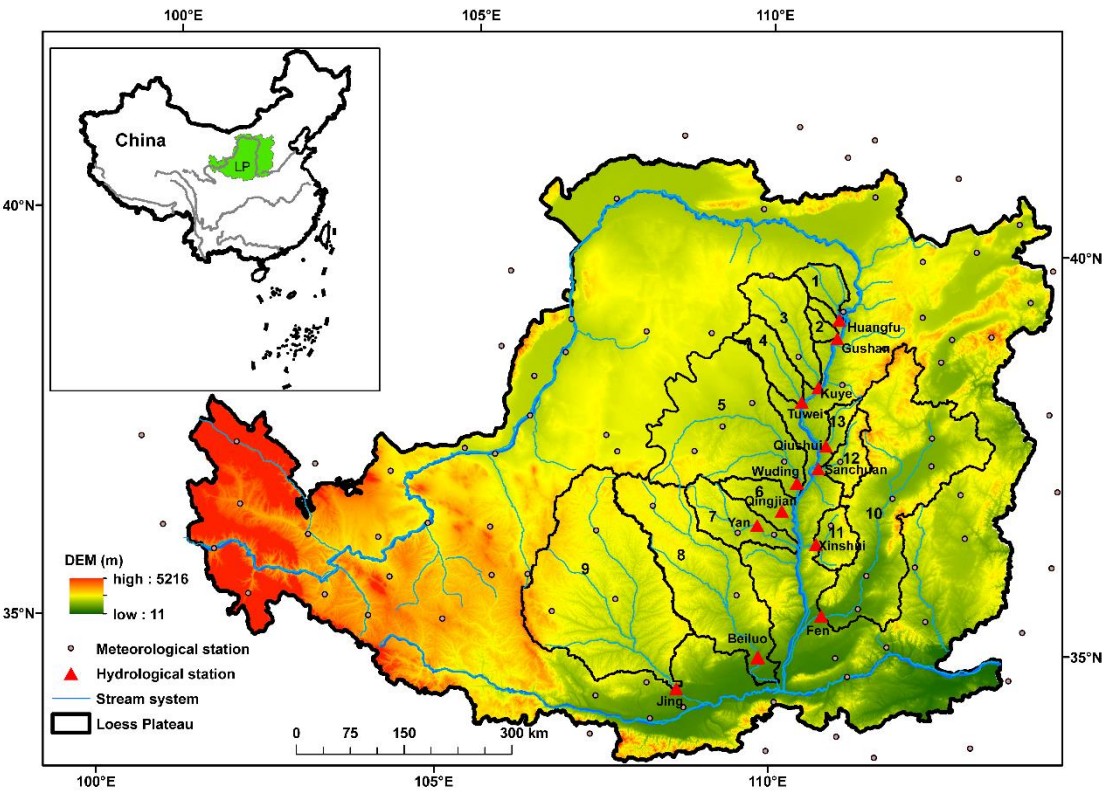

Figure 1. Locations of the study area and hydrometeorological stations.

Monthly runoff data for the 13 catchments were supplied by the Yellow River Conservancy Commission. Detailed information about the catchment characteristics and data durations are shown in Table 1. Daily meteorological data (1960–2012) comprised of precipitation, daily maximum and minimum temperatures, atmospheric pressures, wind speeds, mean relative humidity values, and sunshine durations, which were recorded at 96 stations, were provided by the China Meteorological Administration. The new NDVI third generation (NDVI3g) dataset was used to represent the vegetation characteristics of the study area, and detailed information about this dataset was presented earlier by



Fensholt and Proud (2012). The maximum value compositing (MVC) procedure (Holben, 1986) was
applied to produce the annual NDVI values.
Table 1. Long-term hydrometeorological characteristics and vegetation coverage (1981–2012)[a].

| ID | Basin | Data length, year | $P$, mm/yr | $ET_0$, mm/yr | $ET$, mm/yr | $\omega$ | $M$ | $S$ |
|---|---|---|---|---|---|---|---|---|
| 1 | Huangfu | 32 | 372 | 972 | 347 | 3.15 | 0.42 | 0.94 |
| 2 | Gushan | 25 | 394 | 975 | 359 | 2.75 | 0.45 | 0.79 |
| 3 | Kuye | 32 | 375 | 1018 | 333 | 2.45 | 0.43 | 0.99 |
| 4 | Tuwei | 32 | 383 | 1031 | 308 | 1.99 | 0.41 | 0.95 |
| 5 | Wuding | 32 | 385 | 1045 | 356 | 2.68 | 0.46 | 0.95 |
| 6 | Qingjian | 32 | 451 | 1009 | 417 | 3.00 | 0.60 | 0.60 |
| 7 | Yan | 32 | 462 | 984 | 433 | 3.21 | 0.70 | 0.51 |
| 8 | Beiluo | 28 | 502 | 960 | 475 | 3.76 | 0.88 | 0.34 |
| 9 | Jing | 32 | 529 | 936 | 497 | 3.74 | 0.59 | 0.51 |
| 10 | Fen | 29 | 465 | 982 | 452 | 4.21 | 0.87 | 0.43 |
| 11 | Xinshui | 32 | 478 | 992 | 458 | 3.77 | 0.87 | 0.45 |
| 12 | Sanchuan | 24 | 444 | 998 | 397 | 2.70 | 0.57 | 0.58 |
| 13 | Qiushui | 23 | 442 | 1006 | 418 | 3.33 | 0.67 | 0.60 |

[a]Because a few runoff data points were missing for several basins, the data length in these basins was less than 32. Each item represents the mean annual value.

## 182   4. Results

### 183   4.1. Interannual variability of parameter $\omega$

The Budyko framework is usually used for analyses of long-term average data on catchment-scale

water balances; however, in this study, it was employed for the interpretation of the interannual
variability of the water balances by using the hydrological year approach described earlier. To validate
the feasibility of using Fu's equation for interannual variability, the evapotranspiration ratio ($ET/P$) and
dryness index ($ET_0/P$) on an annual scale for 13 basins are presented in the supporting information
(Figure S1), and it can be seen that almost all points are focused on Fu's curves in each basin. Therefore,
Fu's equation was considered adequate for the analysis of the interannual variability of the water



balance.

If the controlling parameter $\omega$ on an annual scale can reflect the combined impacts of vegetation change and climate seasonality, it should also exhibit interannual variability with the seasonal variation in vegetation and climate, especially in those catchments affected significantly by climate change and human activities. Obviously, this is true for basins in Loess Plateau (Figure 2). During 1961–2012, $\omega$ values in all 13 basins had an upward trend. Along with such a changing trend in $\omega$, $ET$ should increased for the same levels of $P$ and $ET_0$. Before the 1980s, the variation in $\omega$ for each basin was relatively gentle; however, since that time, it has increased dramatically.








Figure 2. Interannual variability of parameter $\omega$ for 13 basins during 1961 to 2012.

## 4.2. Development of the semi-empirical formula for parameter $\omega$

The relationships between the annual parameter $\omega$ and vegetation coverage $M$ as well as the





climate seasonality index $S$ were first explored in each study basin during the period 1981–2012, and
the results are shown in Figures 3 and 4. We can see that the parameter $\omega$ generally had a positive
correlation with $M$, which implies that evapotranspiration increased with improvements in the
vegetation conditions. However, $\omega$ was correlated negatively with $S$, which means that larger seasonal
variations of coupled water and energy resulted in less evapotranspiration in this area. The relationships
between $\omega$ and $M$ as well as $S$ imply that the annual variation in parameter $\omega$ can be estimated by the
changes in vegetation dynamics and climate seasonality.






Figure 3. Relationships between the annual $\omega$ and vegetation coverage for each basin.





Figure 4. Relationships between the annual $\omega$ and climate seasonality index for each basin.

To expand the sample size and span a wider range of climate conditions, as well as to make the





derived semi-empirical formula of parameter $\omega$ more representative, relationships were then developed
based on the combined dataset from the 13 basins (Figure 5). These results also indicate a good
relationship between $\omega$ and $M$ ($R^2 = 0.40$, $p < 0.01$) as well as $S$ ($R^2 = 0.28$, $p < 0.01$).

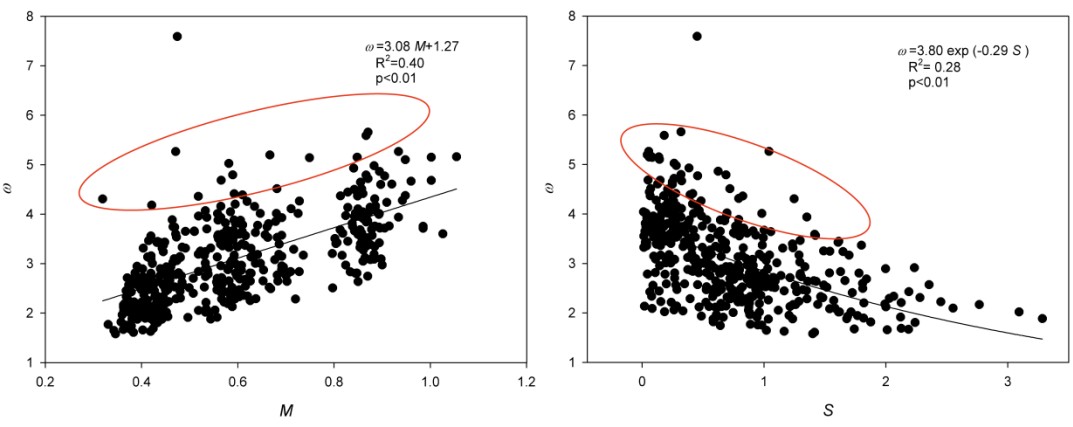


Figure 5. Relationships between the (a) annual $\omega$ and vegetation coverage ($M$) and (b) $\omega$ and climate seasonality
index ($S$) based on the combined dataset from 13 basins.
To develop the semi-empirical formula of parameter $\omega$, the limiting conditions of the two
variables were considered as follows:
(1) If $M \to 0$, i.e. the land surface was bare, which indicates that the climate was extremely dry,

$P \to 0$, $ET \to 0$, and thus, $\omega \to 1$;

(2) If $S \to \infty$, i.e. $P$ and $ET_0$ were completely out of phase, which means the total precipitation

within a year was all concentrated within a particular point of time, $R \to P$, $ET \to 0$, and

$\omega \to 1$.

Considering the relationships shown in Figure 5 and given the above limiting conditions, the
general form of parameter $\omega$ can be expressed as follows:
$\omega = 1 + a \times M^b \times \exp(cS)$         (8)
where $a$, $b$, and $c$ are constants. Using the least linear square regression method, the semi-empirical




formula of parameter $\omega$ is derived as follows:

$$\omega = 1 + 3.525 \times M^{0.783} \times \exp(-0.218\,S) \tag{9}$$

The coefficient of determination $R^2$ and the statistics for the F test of the modelled $\omega$ were 0.51 and 218.94, respectively. Thus, incorporation of the climate seasonality index into the semi-empirical formula of parameter $\omega$ improved the estimation of parameter $\omega$ by advancing the determining coefficient ($R^2$) from 0.45 to 0.51 as compared to the formula that only considered the vegetation information (see Table 2).

Table 2. Analysis of factors affecting parameter $\omega$, as determined by the stepwise regression method.

| Variables | $R^2$ | F | Model coefficients | | |
|---|---|---|---|---|---|
| | | | Ln a | b | c |
| *M* | 0.45 | 336.57 | 3.32 | 0.979 | |
| *M,S* | 0.51 | 218.94 | 3.525 | 0.783 | -0.218 |

A cross-validation approach was chosen to calibrate and test the above semi-empirical formula for parameter $\omega$. Specifically, the dataset for the 13 basins in our study was separated into two groups. One was applied to build the semi-empirical formula, and it consisted of 12 basins for each time; the other was used for testing the performance of the semi-empirical formula, and it consisted of the remaining 1 basin. In total, the cross-validation process was conducted 13 times. After building the semi-empirical formula by using the vegetation coverage data and climate seasonality index data for the 12 basins, the parameter $\omega$ for the validated basin was modelled by using this fitted formula, and the annual *ET* for the validated basin was evaluated with the modelled $\omega$, which is referred to as the "modelled" *ET*. Then, the "modelled" *ET* was compared with the "measured" *ET*.

Table 3 shows the cross-validation results for each basin. The model coefficients of each calibration formula for parameter $\omega$ were very close with the coefficients of Eq. (9). Except for the Tuwei and Sanchuan basins, the MAE (mean absolute error) and RMSE (square root of the mean square error) values for each cross-validation process were relative low, with mean values of 13.5 mm and 16.8





mm, respectively. The NSE coefficient (Nash–Sutcliffe coefficient of efficiency) for each process was
greater than 0.8, thus suggesting that vegetation changes and climate seasonality can well explain the
interannual variation in the controlling parameter of the catchment water balance.

Table 3. Cross-validation results for each basin.

| ID | Validated basin | Model coefficients | | | ET estimation accuracy | | |
|----|-----------------|------|-------|--------|------|------|------|
|    |                 | a    | b     | c      | MAE  | RMS  | NSE  |
| 1  | Huangfu         | 3.597 | 0.868 | -0.228 | 22.3 | 23.8 | 0.88 |
| 2  | Gushan          | 3.525 | 0.787 | -0.231 | 16.3 | 21.3 | 0.90 |
| 3  | Kuye            | 3.490 | 0.743 | -0.233 | 17.4 | 22.7 | 0.88 |
| 4  | Tuwei           | 3.350 | 0.627 | -0.224 | 33.4 | 37.5 | 0.84 |
| 5  | Wuding          | 3.525 | 0.803 | -0.211 | 8.3  | 12.5 | 0.97 |
| 6  | Qingjian        | 3.525 | 0.794 | -0.206 | 13.9 | 18.1 | 0.96 |
| 7  | Yan             | 3.560 | 0.803 | -0.210 | 11.3 | 14.0 | 0.98 |
| 8  | Beiluo          | 3.633 | 0.826 | -0.213 | 10.2 | 11.9 | 0.97 |
| 9  | Jing            | 3.456 | 0.814 | -0.188 | 23.1 | 25.8 | 0.87 |
| 10 | Fen             | 3.421 | 0.738 | -0.223 | 6.3  | 8.9  | 0.98 |
| 11 | Xinshui         | 3.560 | 0.803 | -0.216 | 6.6  | 9.0  | 0.99 |
| 12 | Sanchuan        | 3.561 | 0.782 | -0.215 | 25.6 | 31.0 | 0.88 |
| 13 | Qiushui         | 3.525 | 0.800 | -0.204 | 12.5 | 16.4 | 0.96 |

**4.3. Climate seasonality and vegetation coverage contributions to changes in *ET***

The impacts of vegetation changes on *ET* have been widely studied with the Budyko framework by

assuming surface conditions can be represented by the controlling parameter. However, according to the
developed relationships in our study, the controlling parameter is not only related to surface condition
change, but also to climate seasonality. The contributions of changes in climate ($P$, $ET_0$, and $S$) and
vegetation ($M$) to the changing trend of *ET* were thus estimated by using the semi-empirical formula for
parameter $\omega$ in the context of Fu's framework.

Trend analyses of the hydrometeorological variables and vegetation coverage were first conducted

for each basin (Table S1). $ET_0$, $M$, and $S$ in all basins exhibited an upward trend, though the significance




of the trends were different. In accordance with the method described in Section 2.3, the contributions
of vegetation change and climate seasonality to *ET* trends for each basin were calculated (Table 4). The
changing trend in *ET* was dominated by *P* in the Qingjian, Yan, Beiluo, Jing, Fen and Qiushui River
basins. However, in the Hungfu, Gushan, Kuye, Tuwei, Wuding, and Xinshui River basins, the changing
trend in *ET* was mainly controlled by the improvements in vegetation. Except for the Yan River basin,
improved vegetation in the basins made a positive contribution to the *ET* trend, which is consistent with
the results of Feng et al. (2016). *ET* in several basins showed a downward trend even though *M* made a
positive contribution to the *ET* changes; this was because of the offsetting effect of the other three
factors.
It should be noted that the climate seasonality (represented by *S*) played an important role in the
catchment *ET* variation. The contributions of *S* to *ET* changes ranged from 0.1% to 65.6% (absolute
values). Besides the Gushan, Yan, Fen and Sanchuan River basins, the climate seasonality had a
negative effect on *ET* variation in most of the basins, which means that larger seasonality differences
between seasonal water and heat will lead to smaller amounts of evapotranspiration. Accordingly, if
ω is supposed to only represent the landscape condition, the effects of landscape condition change on
*ET* variation will be underestimated in the Huangfu, Kuye, Tuwei, Wuding, Yan, Jing and Xinshui River
basins, while its effects will be overestimated in the other basins, and the error would be equal to the
contributions of *S* to *ET* changes.
Table 4. Relative contributions of vegetation change and climate seasonality to *ET* trends for each basin[c].

| ID | Basin | $ET_0$, % | P, % | M, % | S, % |
|---|---|---|---|---|---|
| 1 | Huangfu | 8.7 | 30.3 | 61.0 | -0.1 |
| 2 | Gushan | 3.7 | 24.0 | 68.8 | 3.4 |
| 3 | Kuye | 6.4 | 16.4 | 80.3 | -3.2 |
| 4 | Tuwei | 10.3 | 21.7 | 86.5 | -18.4 |
| 5 | Wuding | 4.3 | 31.3 | 78.7 | -14.3 |
| 6 | Qingjian | -3.0 | 72.0 | -15.8 | 46.8 |
| 7 | Yan | -6.5 | 101.6 | 9.6 | -4.7 |





| 8 | Beiluo | -7.5 | 94.0 | -0.4 | 13.8 |
| 9 | Jing | -9.4 | 141.9 | -47.4 | 14.2 |
| 10 | Fen | -10.2 | 158.6 | -18.9 | -29.5 |
| 11 | Xinshui | 22.9 | -29.6 | 133.7 | -27.0 |
| 12 | Sanchuan | -2.7 | 4.2 | 32.8 | 65.6 |
| 13 | Qiushui | -4.8 | 103.1 | -0.1 | 1.7 |

[c]The relative contribution of a certain variable to the *ET* trend ($\varphi(x)$) was calculated as follows: $\varphi(x) = (C\_(x)/C\_(sum)) \times 100\%$, where $C\_(x)$ represents the
contribution of each variable, and $C\_(sum)$ is the sum of the contributions of the four variables.

## 5. Discussion

Although the controlling parameter $\omega$ showed a good relationship with the vegetation change and climate seasonality index, two groups of deviations around the regressed curves were detected (Figure 5). The deviation points for the relationship between $\omega$ and *M* were mainly located at the top of the curve, i.e. corresponding to the same *M* values, where $\omega$ values were greater. We checked those points and found that precipitation and vegetation coverage in those years were normal, but runoff was very low compared to normal years. Excluding abrupt climate change, possible reasons for the extremely low runoff in those years include dam and reservoir operations, as well as irrigation diversions. A study conducted by Liang et al. (2015) on the same basins that we investigated in the Loess Plateau showed that check-dams increased continuously starting from the 1960s. By the year 2006, the numbers of dams along the Fen River and Wuding River reached up to 482 and 181, respectively. Dams can intercept stormwater runoff for a short period during flood seasons and allow more time for infiltration (Polyakov et al., 2014). A total of 21 large and 136 medium-sized reservoirs were installed along the Yellow River by 2001. Such infrastructure can also influence the runoff change by controlling the flooding, regulating the water discharge, and diverting the water to other regions (Chen et al., 2005). Agricultural production is heavily dependent on irrigation throughout the entire Yellow River basin, and it has been reported that water consumption by agricultural irrigation accounted for nearly 80.0% of the entire water consumed





from 1998 to 2011 (Wang et al., 2014). Thereby, water withdrawn for irrigation also plays an important role in the changing trends in runoff. In this study, the deviation points around the relationship curve between the annual $\omega$ and $S$ fell in the upper left, and they were likely influenced by the low runoff. However, separation of the impacts on runoff from vegetation change, climate seasonality, and engineering works will have to await future work.

The relationships of parameter $\omega$ with vegetation dynamics and climate seasonality in some single basins were not significant in this study. Similarly, Yang et al. (2014) also found a weak relationship between parameter $n$ and vegetation coverage in 201 basins in China. This implies that the parameter might represent the combined effects of some other factors. For example, strong interactions among vegetation, climate, and soil conditions will lead to specific hydrologic partitioning at the catchment scale. In dry years, with low soil water contents, plants are trying to adapt by making use of hydrological processes, e.g. ground water dynamics and plant water storage mechanisms, etc. (Renger and Wessolek, 2010). Therefore, the relationship between the parameter and vegetation dynamics can be influenced by climate and soil conditions. However, it is difficult to separate the climatic and soil components from the vegetation change. Moreover, Zhang et al. (2001) reported that the impact of different vegetation types on catchment water balance can be vastly different, and the plant-available water coefficient in their function, which is similar to parameter $\omega$ in Fu's equation, is related to vegetation type. Therefore, the vegetation type may also be an important variable that influences the parameter $\omega$.

In previous attribution analyses of water balance variation based on the climate elasticity method under the Budyko framework (e.g. (Roderick and Farquhar, 2011; Xu et al., 2014; Liang et al., 2015)), the study period was first divided into two periods based on the breakpoint. Then, the effect of a certain variable (i.e. $ET_0$, $P$, or $\omega$) on the change in annual runoff ($R$) or $ET$ from the first period to second period was defined as the product of the elasticity coefficient (i.e. $\varepsilon_P, \varepsilon_{ET_0},$ or $\varepsilon_\omega$) and the variation in



the variable for the two periods (i.e. $\Delta P, \Delta ET_0$, or $\Delta \omega$). This method can be successfully applied to the attribution analysis of the $R$ or $ET$ variation when the breakpoints of annual $R$ or $ET$ are significant. However, if such a breakpoint does not exist, or if there is more than one breakpoint for the time series of annual $R$ or $ET$, the uncertainty of this method will be amplified. For example, even if the elasticity coefficients are constant for any sub-period, the $\Delta P, \Delta ET_0$, and $\Delta \omega$ will change with the different breakpoints. As a result, the contributions of each variable to $R$ or $ET$ will change as well. In our study, the variations in $R$ or $ET$ over the whole period were first distributed equally to every year, i.e. the change rate of $R$ or $ET$ per year, and then, the attribution analysis was conducted for the average annual variation in $R$ or $ET$. This two-part method should yield similar results because the total increment of $R$ or $ET$ for the whole study period in our study, i.e. the product of the change rate and study period, is theoretically close to the variation from the two sub-periods in the above method. Take the Kuye River in our study as an example; the relative contributions of $P$, $ET$, and parameter $\omega$ to the $ET$ variation were 7%, 7%, and 84%, respectively, by using the above previously applied method, while these values with our proposed method were 16%, 6%, and 77%, respectively. The errors mainly were induced by the insignificant breakpoint of the ET series. Thus, the method we used should more applicable and the attribution results more stable because it will not be influenced by the choice of the breakpoint.

## 6. Conclusions

This study explored the concomitant effects of vegetation dynamics and climate seasonality on the variation in interannual controlling parameter $\omega$ from Fu's equation within the Loess Plateau. First, to reduce the impact of ignoring the water storage change on annual catchment water balance, the hydrological year approach was introduced to examine the interannual variability of the controlling parameter $\omega$ for the 13 basins in the Loess Plateau from 1961 to 2012. The findings showed that parameter $\omega$ in all these basins presented an increasing trend, especially after the 1980s. Furthermore,



we checked the relationship between $\omega$ and vegetation dynamics (represented by the annual vegetation coverage, *M*) as well as climate seasonality (represented by the climate seasonality index, *S*). The interannual changes of parameter $\omega$ were found to be related strongly to *M* and *S*. As such, a semi-empirical formula for the annual value of $\omega$ was developed based on these two parameters, and it was proven superior for estimating the actual evapotranspiration (*ET*) by a cross-validation approach. Finally, based on the proposed semi-empirical formula for parameter $\omega$, the contributions of changes in climate (*P*, $ET_0$, and *S*) and vegetation (*M*) to *ET* variations were estimated. The results showed that the improved vegetation conditions in all basins made a positive contribution to the *ET* trend, but these effects were largely offset by other variables in some basins. The contribution of landscape condition changes to ET variation will be estimated with a large error if the effects of climate seasonality were ignored.

## Acknowledgments

This study was supported by the National Key Research and Development Program of China (No.2016YFC0501602), the National Natural Science Foundation of China (No. 41571036), and the Public Welfare Industry (Meteorological) Research Project of China (No. GYHY201506001).

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
