# Peer review of "Vegetation dynamics and climate seasonality jointly control the interannual"

_Hydrology and Earth System Sciences, 2016_

## Referee Comment (RC1) · Anonymous Referee #1 · 30 Oct 2016

Budyko hypothesis, as an important tool to study water and energy balance in catchments, has been developed and enriched in the past few decades. In recent years, it regains researchers' attention as a necessary to examine the changes in catchment landscapes and hydrological conditions given the impacts from climate change and intensified anthropogenic activities. Based on the previous researches - interpreting basin water balance by relating either to vegetation dynamics or to climate seasonality, this article considers both as key factors to drive water balance variations, and comes up with a new two-factors-coupled parameterization scheme for the index 'oumiga' in the Fu's equation under Budyko framework. The innovation of this work lying at providing an integrated angle to estimate regional water balance.

General comments: 1. Usually, the Budyko framework is used in long-term scale so that the water storage change can be ignored. It is a big challenge to apply this framework in interannual catchment water balance. The hydrological year is better than the calendar year, but it is not enough. In present researches, the parameter, such as oumiga, was determined for each catchments then the relation between this parameter and other factors, such vegetation, landscape and climate characteristics were discussed. For example, Li et al. published in WRR in 2013. Therefore, my advice is to set up the relation between oumiga and M and S based on the long-term water balance for the 13 catchments then to discuss the contribution of different part to the runoff change. 2. For the contribution analysis, it is better to divide the whole period into two periods, for example, before 1980 and after 1980. A, B and C in Equation 5 can be estimated by the P, ET0 and oumiga of the whole period. Then deltaP=P2-P1, deltaET0=ET02-ET01, delta oumiga = oumiga 2- oumiga 1. After that, the contribution of P, ET0 and oumiga can be estimated. 3. It is better to analyze the trend of runoff and climate factors with MK test.

Special comments: 1. Please give more detail about the climate seasonality index (S). 2. L225, "out of phase", "out phase", which one is right? 3. L237, just from 0.45 to 0.51, it is not a significant improvement. 4. L242, crossing-validation is not a good choice here because each catchments has its own characteristics, so it can not be validated by other catchments. 5. keep all the panels (including the label, range and scale of x/y-axis) within a figure be consistent. Have a close look at the Fig 2-4. 6. It may be better to replace Fig 3 and 4 by a table show R2 with a certain category. The original figures could be provided as supplementary documents. 7. Table 4, "Relative contributions of vegetation change and climate seasonality to ET trends for each basin", which miss out the contributors from "ET0 and P". 8. For reading convenience, better to insert the ordering number according to the ordering system given in Fig1 and Table1 in the text when mentioning a particular basin in Results.

---

## Referee Comment (RC2) · Anonymous Referee #2 · 1 Dec 2016

This paper evaluates the dependence of Budyko parameter w on vegetation coverage and the climate seasonality on inter-annual timescale. It is interesting to quantitatively estimate the contribution of intra-annual climate variability to annual water balance factors. This manuscript is well written and falls within the scope of HESS. However, some revisions are required, which are given as follows:

The major concern is that the inter-annual water storage change is assumed to be negligible even though hydrologic year is used. The estimated value of w could be affected by this assumption of storage change. Since the purpose of this study is to evaluate

the contribution of vegetation and seasonal climate variability to inter-annual variability of water balance, this assumption is important and needs to be further invsetigated and discussed.

To develop the semi-empirical formula of parameter w, the limiting conditions of M and S were considered in this paper, which is significant for understanding the variability of water balance under the extremely hydrometeorological conditions. However, I think the limiting condition of S is not exactly right: when $\hat{L}\hat{E}\to\infty$ and $\delta\_{(ãĂŰ ET ãĂŮ\_0 )}\neq 0$ in the equation (3), i.e. P→0, and monthly ET0 is not uniform distributed within a year, w can also be close to unity.

It has been reported that the first-order approximation (ignoring the higher orders of the Taylor expansion) in the Equations (4-6) will bring errors (Yang et al. 2014, WRR); furthermore, the function of P and ET0, and their interaction may play some roles in the attribution analysis. Thus, it is better to consider these errors in the paper.

In previous attribution analyses of variation in runoff or ET based on the climate elasticity method, the study period was first divided into two periods, and then the contribution of a variable on the change in runoff or ET from the first period to second period was defined as the product of the elasticity coefficient and the variation of this variable. While in this study, the climate elasticity method was used to explain the change trend of ET for the whole study period. Even the comparisons of these two methods was conducted in the discussion section, there still need more data to support this estimation.

Line 64. Also cite Donohue et al. 2012 JOH.

---

## Short Comment (SC1) · 1 Dec 2016

This paper investigated the influence factors on the annual trend of ET in the LP, and present a equation to calculate the parameter in Budyko frame. Generally, this paper is well written and easy to follow. The equation between parameter w and M, S presented in this study is interesting. I believe the data and results are solid and reasonable. My main concerns are about the discussion section. It would be better to add some discussions on the uncertainties of the method in this study:

In the attribution equation (6), the impact factors are precipitation, ET0 and w. In equation (7), the authors further present that w is the function of M and S. S is a function of precipitation and ET0. Thus, in equation (6), precipitation, ET0 and w are not independent. This independence could have impacts on partial derivatives. This uncertainties could be added and presented in the paper.

Additionally, the impacts of interannual changes of water storage could also be discussed in the paper. The traditional Budyko frame, i.e., equation (1), is conducted on average annual timescale. Therefore, delt_s can be ignored. In this study, the timescale is interannual and delts_s could be discussed in the uncertainty section. It would be better to add some reference to show that delts_s can be ignored on interannual timescale in the LP. The LP is a sub-arid and sub-humid area and delts_s may be relative small on interannual timescale.

---

## Author Comment (AC1) · 4 Jan 2017

Thank you very much for the constructive suggestions. We have revised the manuscript according to the comments. Please check the following response for detailed modification.

1. *In the attribution equation (6), the impact factors are precipitation, ET0 and w. In equation (7), the authors further present that w is the function of M and S. S is a function of precipitation and ET0. Thus, in equation (6), precipitation, ET0 and w are not independent. This independence could have impacts on partial derivatives. This uncertainties could be added and presented in the paper.*

*Response*: Thanks for your comments. However, it must be noted that the concepts of "$P$" and "$ET_0$" in the equation (6) and (3) are different: in the equation (6), they refer to the total precipitation and potential evapotranspiration in a year, i.e. annual $P$ and $ET_0$; while in the equation (3), they represent the intra-annual distribution characteristics of precipitation and potential evapotranspiration, respectively, and thus the information of annual total amounts does not been contained in this equation. Therefore, $P$, $ET_0$ and $\omega$ are independent. The uncertainties in the contribution quantification are mainly from $\omega$ interpretation. The residuals in this study suggest that $\omega$ cannot be fully explained by M and S, and more factors should be incorporated.

As the changes in evapotranspiration has been decomposed without residual by the complementary method (Equation 6-7), the errors were mainly induced by the developed empirical formula for $\omega$ (Equation 11). It suggested that $\omega$ cannot be completely explained by *M* and *S*, and it might include some other factors. Therefore, discussing more factors influencing $\omega$ remains future work.

2. *The impacts of interannual changes of water storage could also be discussed in the paper. The traditional Budyko frame, i.e., equation (1), is conducted on average annual timescale. Therefore, delt_s can be ignored. In this study, the timescale is interannual and delts_s could be discussed in the uncertainty section. It would be better to add some reference to show that delts_s can be ignored on interannual timescale in the LP. The LP is a sub-arid and sub-humid area and delts_s may be relative small on interannual timescale.*

*Response*: We very agree with your opinion. In the section of discussion, we have added some reference according to your suggestion to show that the water storage change in the Loess Plateau is relative small compared with other regions of China.

Despite that catchment-scale water storage changes are usually assumed to be zero on long-term scale, the interannual variability of storage change can be an important component in annual water budget during dry or wet years (Wang and Alimohammadi, 2012), and cannot be ignored. However, the Loess Plateau has a subhumid to semiarid climate, the water storage and its annual variation are relatively small compared with humid regions (see Figure 5 from Mo et al., 2016).

For example, using GRACE (Gravity Recovery and Climate Experiment), the water storage variations in the Yangtze, Yellow and Zhujiang from 2003 to 2008 were analyzed by Zhao et al. (2011), and the values for the Yangtze and Zhujiang basins were 37.8 mm and 65.2 mm, while no clear annual variations are observed in the Yellow River basin (3.0 mm). Furthermore, Mo et al. (2016) found that the water storage in Yellow River kept decreasing from 2004 to 2011, whereas it was changing slowly with a rate of 1.3 mm yr$^{-1}$. Therefore, considering the small water storage change in study area, ignoring water storage change in a period of hydrologic year is reasonable.

References:

Mo, X., Wu, J. J., Wang, Q., and Zhou, H.: Variations in water storage in China over recent decades from GRACE observations and GLDAS, Natural Hazards and Earth System Sciences, 16, 469-482, 10.5194/nhess-16-469-2016, 2016.

Zhao, Q. L., Liu, X. L., Ditmar, P., Siemes, C., Revtova, E., Hashemi-Farahani, H., and Klees, R.: Water storage variations of the Yangtze, Yellow, and Zhujiang river basins derived from the DEOS Mass Transport (DMT-1) model, Science China-Earth Sciences, 54, 667-677, 10.1007/s11430-010-4096-7, 2011.

---

## Author Comment (AC2) · 4 Jan 2017

We greatly appreciate the constructive suggestions and have carefully revised the manuscript accordingly. Please check the following responses for our detailed modification.

**Point-to-point responses:**

Anonymous Referee #1

General comments:

1. *Usually, the Budyko framework is used in long-term scale so that the water storage change can be ignored. It is a big challenge to apply this framework in interannual catchment water balance. The hydrological year is better than the calendar year, but it is not enough. In present researches, the parameter, such as oumiga, was determined for each catchments then the relation between this parameter and other factors, such vegetation, landscape and climate characteristics were discussed. For example, Li et al. published in WRR in 2013. Therefore, my advice is to set up the relation between oumiga and M and S based on the long-term water balance for the 13 catchments then to discuss the contribution of different part to the runoff change.*

*Response*: Thanks for your good comments. We agree your opinion that the Budyko framework is mostly used in long-term scale. The reason why we use a period of hydrologic year to develop the semi-empirical formula is to exclude the cross correlation between $M$ and $S$. We checked the relationship of $M$ and $S$ on the 5-year, 10-year and 30-year scales, and found that they are cross correlated. Particularly, the correlation coefficients increase with the lengths of time scale, and the determining coefficient ($R^2$) is 0.8 for the 30-year scale. If $M$ and $S$ is not independent of each other, they cannot be used together to express different functions. After substantial tests, we found that the relationship between $M$ and $S$ is not significant in a period of hydrologic year. Thus, they can be used to express the controlling parameter. Furthermore,there are several researches have figured out that although the parameter in the Budyko relationship has been used to represent the catchment characteristics, this parameter is also affected by climate seasonality (Milly, 1994;Donohue et al., 2011;Williams et al., 2012;Berghuijs and Woods, 2016;Zhou et al., 2016). In our study, we also found that parameter $\omega$ has a negative correction with climate seasonality. Thus, the climate seasonality should be incorporating into the parameter.

Your concern about the changes in water storage is really very important in water balance equation. To exclude the potential impacts, we checked the interannual variation of catchment-scale water storage in some studies, and presented some discussion about this part. Specifically, we found that the water storage change in the Loess Plateau is relative small compared with the other regions of China. In such cases, assessing catchment-scale water balance by ignoring water storage change should be reasonable on a time scale of hydrologic year.

Despite that catchment-scale water storage changes are usually assumed to be

zero on long-term scale, the interannual variability of storage change can be an important component in annual water budget during dry or wet years (Wang and Alimohammadi, 2012), and cannot be ignored. However, the Loess Plateau has a subhumid to semiarid climate, the water storage and its annual variation are relatively small compared with humid regions (see Figure 5 from Mo et al., 2016). For example, using GRACE (Gravity Recovery and Climate Experiment), the water storage variations in the Yangtze, Yellow and Zhujiang from 2003 to 2008 were analyzed by Zhao et al. (2011), and the values for the Yangtze and Zhujiang basins were 37.8 mm and 65.2 mm, while no clear annual variations are observed in the Yellow River basin (3.0 mm). Furthermore, Mo et al. (2016) found that the water storage in Yellow River kept decreasing from 2004 to 2011, whereas it was changing slowly with a rate of 1.3 mm yr$^{-1}$. Therefore, considering the small water storage change in study area, ignoring water storage change in a period of hydrologic year is reasonable.

2. *For the contribution analysis, it is better to divide the whole period into two periods, for example, before 1980 and after 1980. A, B and C in Equation 5 can be estimated by the P, ET0 and oumiga of the whole period. Then deltaP=P2-P1, deltaET0=ET02-ET01, delta oumiga = oumiga 2- oumiga 1. After that, the contribution of P, ET0 and oumiga can be estimated.*

Response: Thanks for reviewer's good suggestion. We recalculated the contribution according to your suggestion by dividing the whole study period into two subperiods. Further, we replaced the previous method with a new method developed by Zhou et al. (2016). The previous method ignored the higher orders of the Taylor expansion and resulted in errors; however, the new method proposed by Zhou et al. (2016) decompose the runoff/ET changes into two components precisely without any residuals. And the detailed revisions are showed in the section 2.3 and 4.3.

[revised manuscript text omitted]

*3. It is better to analyze the trend of runoff and climate factors with MK test.*

*Response*: The trend analysis of each variable has been shown in table 4.

Special comments:

*1. Please give more detail about the climate seasonality index (S).*

*Response*: Thanks for reviewer's good suggestion, and the more detailed description has been added in the section 2.2.

Solar radiation was considered as the dominant factor that controls the climate seasonality and thus the seasonality of $P$ and $ET_0$ can be can be expressed by sine functions (Milly, 1994;Woods, 2003):

$$P(t) = \bar{P}(1 + \delta_P \sin\omega t) \tag{3a}$$

$$ET_0(t) = \overline{ET_0}(1 + \delta_{ET_0} \sin\omega t) \tag{3b}$$

where $\bar{P}$ and $\overline{ET_0}$ are the mean monthly $P$ and $ET_0$; $\delta_P$ and $\delta_{ET_0}$ are the seasonal amplitude of precipitation and potential evapotranspiration, respectively. The values of $\delta_P$ and $\delta_{ET_0}$ might both range from -1 to 1 because $P$ and $ET_0$ always have positive value on physical grounds. Larger absolute values of $\delta_P$ and $\delta_{ET_0}$ mean larger variability of climate seasonality. $\varphi$ is the duration of the seasonal cycle, $2\pi\varphi$ equal to 1 year. Woods (2003) summarized the modelled climate of Eqs.(3a) and (3b) in dimensionless form and defined the climate seasonality index ($S$) and here it was used to reflect the non-uniformity in the annual distribution of water and heat in our study:

$$S = \left|\delta_P - \delta_{ET_0}\emptyset\right| \tag{4}$$

where $\emptyset$ is the dryness index, $\emptyset = \overline{ET_0}/\bar{P}$. If $S=0$, there is no seasonal fluctuation of the difference between $P$ and $ET_0$. Larger values of $S$ indicate that the larger changes in the balance between $P$ and $ET_0$ during the seasonal cycle.

*2. L225, "out of phase", "out phase", which one is right?*

*Response*: "out of phase" should be more suitable. The same expression was also used by Potter et al. in WRR in 2005.

*3. L237, just from 0.45 to 0.51, it is not a significant improvement.*

*Response*: Thanks for reviewer's comment. The more important meaning of

incorporating the climate seasonality into the controlling parameter oumiga is to further explore the factors that controlling the interannual catchment water balance, rather than only considering its function of improving the estimation of parameter oumiga. And the results of attribution analysis showed that the contribution of vegetation coverage changes to ET variation will be estimated with a large error if the effects of climate seasonality were ignored. Thus, we will no longer address the improvement of the estimation of parameter oumiga after considering climate seasonality, and remove Table 2 as well as related comparison.

4. *L242, crossing-validation is not a good choice here because each catchments has its own characteristics, so it can not be validated by other catchments.*

*Response*: We agree. Each catchment has its own characteristics, mainly including the underlying physical conditions (such as soil properties and topography), vegetation and climate characteristics. Ignoring the spatial heterogeneity of underlying physical conditions for studied basins may influence the performance of the empirical equation we built. However, we think that the crossing-validation approach can be used to calibrate and test the semi-empirical formula for parameter oumiga, because the rotated calibrations using 12 basins instead of all 13 basins only produce slight variations in the slopes and intercepts from regressions (Table 3), which suggests that the formula we built are robust and it can be used to assess catchment actual evapotranspiration in the Loess Plateau. And the subsequent validations further prove the good performance of our formula. This method was also widely used by previous studies (e.g. Li et al., 2013;Chen et al., 2014;Schnier and Cai, 2014;Kim et al., 2015;Nerini et al., 2015;Lv and Zhou, 2016;Rakovec et al., 2016;Toth, 2016). Among them, the study of Li et al. (2013) is similar with ours, who also used the crossing-validation approach to test their formula for each catchment.

5. *Keep all the panels (including the label, range and scale of x/yaxis) within a figure be consistent. Have a close look at the Fig 2-4.*

*Response*: Thanks for reviewer's suggestion, and we have unified the format of Fig 2-4 & S1.

6. *It may be better to replace Fig 3 and 4 by a table show $R^2$ with a certain category. The original figures could be provided as supplementary documents.*

*Response*: We have moved the Fig3 &4 into the supplementary documents. Since these two figures have contained "$R^2$" and "p", we think it is not necessary to make new tables.

7. *Table 4, "Relative contributions of vegetation change and climate seasonality to ET trends for each basin", which miss out the contributors from "ET0 and P".*

*Response*: We have corrected this title as "Attribution analysis for *ET* changes for each basin"

8. *For reading convenience, better to insert the ordering number according to the ordering system given in Fig1 and Table1 in the text when mentioning a particular basin in Results.*

*Response*: We have inserted the ordering number of each basin in the revised text, for example, "Huangfu" was revised as "basin #1".

---

## Author Comment (AC3) · 4 Jan 2017

We are greatly thankful for the insightful and constructive comments from the anonymous reviewer. We have carefully studied them and revised the manuscript accordingly. This document contains our specific responses to the comments.

Anonymous Referee #2

1. *The major concern is that the inter-annual water storage change is assumed to be negligible even though hydrologic year is used. The estimated value of w could be affected by this assumption of storage change. Since the purpose of this study is to evaluate the contribution of vegetation and seasonal climate variability to inter-annual variability of water balance, this assumption is important and needs to be further invsetigated and discussed.*

*Response*: Thanks for reviewer's good suggestion. And the other anonymous referees also figured out this problem and suggested us build the relationship between $\omega$ and $M$ as well as $S$ on the long-term scale. However, after checking the relationship of $M$ and $S$ on the 5-year, 10-year and 30-year scales, we found that they are cross correlated. Particularly, the correlation coefficients increase with the lengths of time scale, and the determining coefficient ($R^2$) is 0.8 for the 30-year scale. If $M$ and $S$ is not independent of each other, they cannot be used together to express different functions. After substantial tests, we found that the relationship between $M$ and $S$ is not significant in a period of hydrologic year. Thus, they can be used to express the controlling parameter. Furthermore,there are several researches have figured out that although the parameter in the Budyko relationship has been used to represent the catchment characteristics, this parameter is also affected by climate seasonality (Milly, 1994;Donohue et al., 2011;Williams et al., 2012;Berghuijs and Woods, 2016;Zhou et al., 2016). In our study, we also found that parameter $\omega$ has a negative correction with climate seasonality. Thus, the climate seasonality should be incorporating into the parameter.

Your concern about the changes in water storage is really very important in water balance equation. To exclude the potential impacts, we checked the interannual variation of catchment-scale water storage in some studies, and presented some discussion about this part. Specifically, we found that the water storage change in the Loess Plateau is relative small compared with the other regions of China. In such cases, assessing catchment-scale water balance by ignoring water storage change should be reasonable on a time scale of hydrologic year.

Despite that catchment-scale water storage changes are usually assumed to be zero on long-term scale, the interannual variability of storage change can be an important component in annual water budget during dry or wet years (Wang and Alimohammadi, 2012), and cannot be ignored. However, the Loess Plateau has a subhumid to semiarid climate, the water storage and its annual variation are relatively small compared with humid regions (see Figure 5 from Mo et al., 2016). For example, using GRACE (Gravity Recovery and Climate Experiment), the water storage variations in the Yangtze, Yellow and Zhujiang from 2003 to 2008 were analyzed by Zhao et al. (2011), and the values for the Yangtze and Zhujiang basins were 37.8 mm

and 65.2 mm, while no clear annual variations are observed in the Yellow River basin (3.0 mm). Furthermore, Mo et al. (2016) found that the water storage in Yellow River kept decreasing from 2004 to 2011, whereas it was changing slowly with a rate of 1.3 mm $yr^{-1}$. Therefore, considering the small water storage change in study area, ignoring water storage change in a period of hydrologic year is reasonable.

2. *To develop the semi-empirical formula of parameter w, the limiting conditions of M and S were considered in this paper, which is significant for understanding the variability of water balance under the extremely hydrometeorological conditions. However, I think the limiting condition of S is not exactly right: when $\emptyset \to \infty$ and $\delta_{ET_0} \neq 0$ in the equation (3), i.e. P→ 0, and monthly ET0 is not uniform distributed within a year, w can also close to unity.*

Response: Yes, the limiting condition of S is indeed not right and we have corrected it according to your suggestion in the revised manuscript, thanks for your carefulness.

If $S \to \infty$, i.e. $\emptyset \to \infty$ and $\delta_{ET_0} \neq 0$ in the equation (3), which means monthly $ET_0$ is not uniform distributed within a year and $P \to 0$, thus $ET \to 0$, and $\omega \to 1$.

3. *It has been reported that the first-order approximation (ignoring the higher orders of the Taylor expansion) in the Equations (4-6) will bring errors (Yang et al. 2014, WRR); furthermore, the function of P and ET0, and their interaction may play some roles in the attribution analysis. Thus, it is better to consider these errors in the paper.*

Response: Thanks for reviewer's good suggestion and we agreed with your opinion. Thus, we applied the new method proposed by Zhou et al. (2016) to conduct attribution analysis. The algebraic identities in their work can ensure that the change in runoff/*ET* can be decomposed into two components precisely without any residuals and reduced the errors of ignoring the higher orders of the Taylor expansion in the traditional attribution method. Furthermore, the errors and uncertainties induced by the attribution analysis have been added and presented in the discussion section.

Errors still exhibited in the attribution analysis of ET changes. As the changes in evapotranspiration has been decomposed without residual by the complementary method (Equation 6-7), the errors were induced from the developed empirical formula for *w* (Equation 11). It suggested that ω cannot be completely explained by M and S, and it might include some other factors. Therefore, discussing more factors influencing ω remains future work.

4. *In previous attribution analyses of variation in runoff or ET based on the climate elasticity method, the study period was first divided into two periods, and then the contribution of a variable on the change in runoff or ET from the first period to second period was defined as the product of the elasticity coefficient and the variation of this variable. While in this study, the climate elasticity method was used to explain the change trend of ET for the whole study period. Even the comparisons of these two methods was conducted in the discussion section, there still need more data to support this estimation.*

*Response*: Thanks for reviewer's good suggestion. We have to admit that the attribution method we used will produce some uncertainties. And other two anonymous referees also figured out this problem. Therefore, considering the suggestions of three referees, "the complementary method" proposed by Zhou et al. (2016) was adopted in our revised manuscript. And the corresponding revision was shown in the section 2.3 and 4.3.

[revised manuscript text omitted]

5. *Line 64. Also cite Donohue et al. 2012 JOH.*

*Response*: This reference has been cited.

Table 5. Attribution analysis for *ET* changes for each basin [c]

| ID | Basin | Break point of ET | Change from Period 1 to Period 2 | | | | | $ET_0$/ $P$/$M$/$S$ induced ET change (mm) | | | | | Contribution to ET change (%) | | | |
|---|---|---|---|---|---|---|---|---|---|---|---|---|---|---|---|---|
| | | | $\Delta ET$ | $\Delta ET_0$ | $\Delta P$ | $\Delta M$ | $\Delta S$ | $C\_$ $(ET_0)$ | $C\_$ $(P)$ | $C\_$ $(\omega)$ | $C\_$ $(M)$ | $C\_$ $(S)$ | $\varphi\_$ $(ET_0)$ | $\varphi\_$ $(P)$ | $\varphi\_$ $(M)$ | $\varphi\_$ $(S)$ |
| 1 | Huangfu | 2001(ns) | 41.7 | 7.0 | 22.2 | 0.03 | 0.01 | 0.28 | 18.67 | 22.70 | 22.73 | -0.04 | 0.7 | 44.8 | 54.6 | -0.1 |
| 2 | Gushan | 2000(ns) | 33.6 | 64.9 | 20.6 | 0.07 | -0.10 | 2.81 | 17.01 | 13.77 | 8.87 | 4.90 | 8.4 | 50.6 | 26.4 | 14.6 |
| 3 | Kuye | 2000(**) | 51.4 | 32.0 | 17.3 | 0.06 | 0.05 | 1.54 | 13.34 | 36.48 | 55.95 | -19.47 | 3.0 | 26.0 | 108.9 | -37.9 |
| 4 | Tuwei | 2000(**) | 43.2 | 39.6 | 24.0 | 0.07 | -0.03 | 2.57 | 15.28 | 25.35 | 21.85 | 3.49 | 5.9 | 35.4 | 50.6 | 8.1 |
| 5 | Wuding | 2000(*) | 35.2 | 17.6 | 26.9 | 0.09 | -0.12 | 0.77 | 21.82 | 12.64 | 8.24 | 4.40 | 2.2 | 61.9 | 23.4 | 12.5 |
| 6 | Qingjian | 1988(**) | -50.1 | 32.0 | -48.0 | 0.08 | 0.19 | 2.06 | -37.80 | -14.31 | -47.09 | 32.78 | -4.1 | 75.5 | 94.08 | -65.5 |
| 7 | Yan | 1985(**) | -82.3 | 44.6 | -86.9 | 0.05 | 0.30 | 3.19 | -69.52 | -15.96 | 22.19 | -38.14 | -3.9 | 84.5 | -27.0 | 46.4 |
| 8 | Beiluo | 1985(**) | -65.1 | 49.4 | -79.8 | 0.01 | 0.19 | 4.33 | -62.9 | -6.75 | 3.69 | -10.43 | -6.6 | 96.3 | -5.7 | 16.0 |
| 9 | Jing | 1990(**) | -33.7 | 43.0 | -47.8 | 0.03 | 0.11 | 4.1 | -37.2 | -0.61 | -8.23 | 7.61 | -12.2 | 110.3 | 24.4 | -22.6 |
| 10 | Fen | 2005(ns) | 23.1 | 8.5 | 21.2 | 0.07 | -0.20 | 0.33 | 19.00 | 3.81 | 2.13 | 1.68 | 1.4 | 82.1 | 9.2 | 7.3 |
| 11 | Xinshui | 1990(**) | -19.1 | 39.7 | -24.7 | 0.02 | 0.09 | 2.06 | -21.08 | -0.14 | 0.41 | -0.55 | -10.8 | 110.1 | -2.1 | 2.9 |
| 12 | Sanchuan | 1996(ns) | -27.0 | 45.4 | -43.4 | -0.01 | 0.22 | 3.01 | -32.52 | 2.56 | 0.20 | 2.36 | -11.2 | 120.6 | -0.7 | -8.8 |
| 13 | Qiushui | 1996(ns) | -80.3 | 77.5 | -103.5 | -0.01 | 0.68 | 3.76 | -83.68 | -0.40 | -0.02 | -0.37 | -4.7 | 104.2 | 0.1 | 0.5 |

[c]The relative contribution of a certain variable to the *ET* change ($\varphi(x)$) was calculated as follows: $\varphi(x) = (C\_(x)/\Delta ET) \times 100\%$, where $C\_(x)$ represents the contribution of each variable.